Neglected Tropical Diseases

# Performance of novel onchocerciasis rapid diagnostic tests in a field setting in South Sudan

Amber Hadermann[1], Stephen Raimon Jada[2], Chiara Trevisan[3], Abdurezak Seid[3], Charlotte Lubbers[1], Luís-Jorge Amaral[1], Marina Saleeb[1], Yak Yak Bol[4], Katja Polman[3], Joseph Nelson Siewe Fodjo[1], Robert Colebunders[1,5]*

1 Global Health Institute, University of Antwerp, Antwerp, Belgium, 2 Amref Health Africa, Juba, Republic of South Sudan, 3 Department of Public Health, Institute of Tropical Medicine, Antwerp, Belgium, 4 Neglected Tropical Diseases Programme, Ministry of Health, Juba, Republic of South Sudan, 5 Department of Tropical Biology, Liverpool School of Tropical Medicine, Liverpool, United Kingdom

* robert.colebunders@lstm.ac.uk

## Abstract

### Introduction

Onchocerciasis is a neglected tropical disease caused by *Onchocerca volvulus* and is primarily controlled through mass drug administration (MDA) of ivermectin. Current diagnostic tests perform sub-optimally for onchocerciasis elimination programs. According to the WHO, a test needs a sensitivity of ≥60% and a specificity of ≥99.8% for mapping, and a sensitivity of ≥89% and a specificity of ≥99.8% for MDA stopping decisions. We assessed the performance of the commercially available Ov16 SD Bioline rapid diagnostic test (RDT) and three novel RDTs: DDTD biplex type A, type C, and the monoplex type GADx.

### Methods

The study included 319 pregnant/post-partum women in Maridi, an onchocerciasis-endemic area in South Sudan. Locally trained healthcare workers conducted RDT with whole blood and skin snip testing. Ov16 SD Bioline RDT was also performed using dried blood spots (DBS). Diagnostic performances were evaluated using three individual reference tests (anti-Ov16 ELISA, O-150 qPCR, skin snip microscopy) and two composite reference standards. DBS were tested for *Mansonella spp.* using qPCR to account for potential false positives.

### Results

Anti-Ov16 ELISA detected *O. volvulus* antibodies in 49.6% (123/248) participants, while *O. volvulus* infection was documented by skin snip microscopy in 31.9% (46/144) participants and in 41.8% (50/141) by qPCR. *Mansonella spp.* infection prevalence was 13.4% (34/253). Monoplex RTDs (Ov16 SD Bioline, GADx) detected

**Data availability statement:** The datasets generated during and/or analysed during the current study are publicly available in the Zenedo repository (DOI: 10.5281/zenodo.17357017).

**Funding:** This work was supported by the Coalition for Operational Research on Neglected Tropical Diseases (COR-NTD), which is funded at The Task Force for Global Health primarily by the Bill & Melinda Gates Foundation (OPP1190754) and by the United States Agency for International Development (USAID) (COR-NTD grant to RC). The study funders had no role in study design, data collection and analysis, decision to publish or preparation of the manuscript.

**Competing interests:** The authors have declared that no competing interests exist.

*O. volvulus* antibodies in >60%, while the biplex tests (DDTD A and C) detected 50% seropositivity. Estimated sensitivities of RDTs ranged from 69.4% (Ov16 SD Bioline) to 48.9% (DDTD A), while specificity ranged from 73.2% (DDTD C) to 42.5% (Ov16 SD Bioline).

## Conclusion

*O. volvulus* and *Mansonella spp.* prevalence was high among pregnant and post-partum women in Maridi. The novel RDTs demonstrated sensitivities within the range of the commercially available Ov16 SD Bioline RDT, meeting the WHO threshold for onchocerciasis mapping but not for MDA stopping decisions. No RDT met the specificity threshold of 99.8%, although some exhibited slightly higher specificity than Ov16 SD Bioline RDT. This suboptimal specificity raises concerns, underscoring the need for better diagnostic tools to support onchocerciasis elimination efforts.

## Author summary

Onchocerciasis, or river blindness, is a parasitic disease transmitted by blackflies and controlled through mass treatment with ivermectin. As countries move toward eliminating the disease, highly accurate diagnostic tools are urgently needed to map infection areas and decide when to stop mass drug administration. In this study, we evaluated the accuracy of four rapid diagnostic tests (RDTs) in detecting infection in pregnant and post-partum women in South Sudan. We compared the RDT results to multiple standard laboratory tests for the parasite, including antibody detection, skin microscopy, and DNA testing. We also tested for co-infection with *Mansonella* parasites, which could interfere with diagnosis. While some new tests performed comparably to the currently available Ov16 SD Bioline RDT, none achieved the high specificity required by the World Health Organization for decisions about stopping treatment programs. Our findings highlight the continued need for improved diagnostic tools that are both sensitive and specific, especially in areas where similar parasitic infections may lead to false-positive results.

## Introduction

Onchocerciasis, commonly referred to as river blindness, is a neglected tropical disease caused by the filarial parasite *Onchocerca volvulus*. The parasite is transmitted to humans through the bites of infected blackflies (*Simulium* sp.) that breed near fast-flowing rivers and streams. The disease is endemic in many regions of sub-Saharan Africa, including Maridi County, South Sudan [1,2]. Control efforts rely primarily on community directed treatment with ivermectin (CDTI), which is sometimes complemented by vector control. In Maridi, vector control is implemented using the

"slash and clear" method, which involves removing all vegetation from the dam spillway, the only blackfly breeding site in the area.

Clinically, onchocerciasis is characterized by a spectrum of symptoms, including dermatological manifestations such as intense itching, papular dermatitis and skin depigmentation, as well as ocular complications that may progress to irreversible blindness if untreated. Onchocerciasis is also associated with epilepsy (onchocerciasis-associated epilepsy, OAE) which includes nodding syndrome, a debilitating condition affecting children in onchocerciasis-endemic regions with high ongoing *O. volvulus* transmission [3,4].

Traditional diagnostic methods, such as skin snip microscopy, have long been considered the gold standard for detecting *O. volvulus* infection. However, this technique has limitations [5]. It is invasive, requiring the excision of a small piece of skin, which often causes discomfort and fear, reducing community acceptance [5]. Additionally, its sensitivity is insufficient in low-prevalence settings, where microfilarial loads are too low for reliable detection [6]. Furthermore, skin snip microscopy is labour-intensive and requires skilled personnel and laboratory infrastructure, making it impractical for large-scale or remote field operations [5]. The invasive nature of the procedure also poses risks of secondary infections, particularly in unsanitary conditions, and raises ethical concerns for use in vulnerable populations, such as pregnant women [5,7]. Moreover, the specificity depends mainly on the skill-level of the laboratory personnel, as differences among microfilaria spp. are minimal [8].

Other reference diagnostic tests, including the anti-Ov16 enzyme-linked immunosorbent assay (ELISA) and O-150 quantitative polymerase chain reaction (qPCR), offer alternative approaches with their own advantages and limitations. The O-150 qPCR detects *O. volvulus* DNA in skin snips with exceptional sensitivity, making it ideal for confirming low-level infections in elimination settings [6]. Nonetheless, its high cost and reliance on advanced molecular laboratory infrastructure restrict its application in field settings. Moreover, the reliability of this test is undermined by the absence of standardization and increasing scientific concerns regarding its non-specificity—issues that remain unaddressed in formal publications.

The anti-Ov16 ELISA detects *O. volvulus*-specific IgG4 antibodies and is highly sensitive, particularly in areas with moderate infection prevalence [9]. However, it requires specialized laboratory equipment and trained personnel, which limits its utility in remote or resource-constrained settings—especially following the discontinuation of the commercially available SD Bioline Ov16 ELISA. To overcome these limitations and enable field-based serological screening, the Ov16 SD Bioline rapid diagnostic test (RDT) was developed. This lateral flow test is among the World Health Organization (WHO) recommended tools for the detection of *O. volvulus* exposure. Mainly valued for its non-invasive nature, suitability for field use, and effectiveness in moderate-to-high transmission settings [6]. However, its performance is insufficient in low-prevalence areas [9]. Additionally, both Ov16 tests have been found to be susceptible to cross-reactivity with other filarial parasites, such as *Loa loa* and *Wuchereria bancrofti*, which can lead to false-positive results in co-endemic regions [10]. Moreover, cross-reactivity to *Mansonella spp.* is a concern which remains insufficiently investigated.

To address these critical gaps in onchocerciasis diagnostics, the WHO introduced two target product profiles (TPPs) in 2021, aimed at guiding the development of new diagnostic tools [11]. The first, 'the mapping TPP', focuses on supporting efforts to map onchocerciasis prevalence in low-transmission (hypo-endemic) areas and determining when to initiate mass drug administration (MDA) of ivermectin. Accurate mapping is crucial in such regions to prevent disease re-emergence. The WHO recommends that diagnostic tools for this purpose should have a sensitivity of ≥60% and a specificity of ≥99.8% [11].

The second, 'the stopping TPP', targets the safe discontinuation of MDA. The required diagnostic sensitivity for stopping MDA is raised to ≥89%, while specificity still needs to be very high ≥99.8% to ensure only true infections are detected [11]. Both TPPs emphasize the need for improved diagnostic tests with increased sensitivity and specificity to support onchocerciasis elimination goals [11].

In response to the WHO's call to enhance the sensitivity and specificity of *O. volvulus* RDTs, Drugs & Diagnostics for Tropical Diseases (DDTD) in San Diego, California, developed a new serological RDT for onchocerciasis [12]. The initial prototype (version A) was designed as a biplex lateral flow assay (LFA) with four antigens distributed across two test lines. Test line 1 (T1) detects human IgG4 antibodies specific to Ov16 and OvOC3261, while test line 2 (T2) detects IgG4 antibodies specific to Ov33.3 [13] and OvOC10469 [14]. The DDTD test is considered positive only if both test lines show a visible result, a criterion that is expected to slightly reduce sensitivity but enhance overall specificity. Due to the presence of two test lines, DDTD RDTs are referred to as biplexes.

In February 2023, a first-generation DDTD biplex A was field-tested in Maridi County, an onchocerciasis-endemic area in Western Equatoria State, South Sudan [15]. The test was found to be user-friendly and practical for field deployment. However, additional research was needed to evaluate its performance [15].

Similarly, the Global Access Diagnostics (GADx) RDT incorporates advanced immunoassay technology and the Ov16 antigen to increase sensitivity and specificity while reducing cross-reactivity. Its robust design ensures minimal dependence on cold chain storage and greater resilience to environmental fluctuations, making it a promising solution for resource-limited settings. The GADx RDT and the Ov16 SD Bioline RDT each contain a single test line and are therefore referred to as monoplex test.

In this study, we compared the performance of three novel RDTs (the DDTD biplex type A and C, and the GADx RDT) with that of the commercially available Ov16 SD Bioline RDT. The evaluation was conducted using three established reference tests (skin snip microscopy, O-150 qPCR and Ov16 ELISA) as well as two composite references. Additionally, we assessed potential cross-reactivity to *Mansonella spp.* This study was designed to contribute to the evidence base required for U.S. Food and Drug Administration (FDA) approval of the novel RDTs. This study was conducted independently under the leadership of the Coalition for Operational Research on Neglected Tropical Diseases (COR-NTD) and funded by the Bill and Melinda Gates Foundation (see acknowledgement) [16].

## Materials and methods

### Ethical approval

Ethical Approval for the study was granted by the Ministry of Health Research Ethics Review Board in Juba, South Sudan (MOH/RERB 56/2022) and the Ethical committee of the University of Antwerp, Belgium (Project ID 3394 - Edge - EudraCT - BUN B3002022000098).

### Informed consent statement

The aims and procedures of the study were explained to all participants in the language of their choice and signed or thumb-printed informed consent was obtained from participants, parents or caregivers, and assent was obtained from adolescents (aged 12–18 years).

### Study setting

The study was conducted in Maridi County, an onchocerciasis-endemic area in Western Equatoria State in South Sudan. Maridi County harbours a population of more than 101,065 inhabitants from five different districts, also referred to as "payams". The main economic and livelihood activity within the area is farming, and the dominant domestic animals raised in the area are goats and sheep, but no pigs [17]. The Maridi dam is the only known breeding site of *Simulium* blackflies in the area. In villages close to the Maridi dam, a high epilepsy prevalence of 4.4% was documented with over 80% of affected individuals meeting the OAE criteria [17]. In addition to onchocerciasis, epilepsy—including nodding syndrome—in South Sudan has also been associated with *M. perstans* infection [18,19].

## Sample collection

From June to September 2023, all pregnant or one-week post-partum women attending the Maridi Hospital (4.915°,29.464°) were asked to participate in the study. Pregnant women represent a valuable population to evaluate RDT performance, as ivermectin remains contra-indicated during pregnancy, making them one of the few remaining groups unaffected by recent ivermectin exposure. As part of the COR-NTD taskforce, this study population will be incorporated into the larger testing cohort.

Eligibility criteria for this study included: i) willingness to participate and provide informed consent, ii) delivering at Maridi Hospital, or visiting the hospital for a pregnancy check-up or child vaccination within one-week post-partum. From all enrolled women, sociodemographic and clinical data were collected by local healthcare workers (HCWs) along with two types of biological samples: dry blood spots (DBS) and skin snips. All HCW received training for all relevant procedures during two sit-down meetings, which covered standard operation procedures for sample collection and diagnostic tests, and ethical consent. Additional personal one-on-one training sessions were provided as needed. Clinical information relevant to *O. volvulus* diagnosis was recorded, including ivermectin intake, presence of pruritus, and the presence of onchocercal nodules. Skin snips were obtained using punch biopsy tools from both sides of the iliac crest. Samples were initially examined on site for *O. volvulus* microfilariae using microscopy, and then preserved in saline for subsequent qPCR analysis in Antwerp, Belgium. Capillary blood obtained by finger prick was collected on Whatman filter paper and air-dried at room temperature. All samples were stored at -20°C prior to shipment to Belgium for further analysis.

## Rapid diagnostic tests

The following RDTs were performed on-site by trained HCWs: the Ov16 RDT (SD Bioline, Inc., Gyeonggi-do, South Korea), DDTD-A, and -C, and GADx. All RDTs were performed following the respective manufacturer's instructions. DDTD RDTs results were considered positive if both test lines T1 and T2 were visible. Test results were recorded in the RedCap data collection tool 30 minutes after addition of the buffer solution at Maridi Hospital. Photographs of the test lines were taken at 20-, 30- and 60-minutes post buffer addition. Photos were taken with the participant identifier (ID) on a separate label and directly on the RDT cassette to ensure optimal traceability. The time of the test administration was recorded and noted on the label. Each photo was time-stamped, by the recording device, allowing for subsequent accuracy verification by the research team in Antwerp, Belgium.

The WHO's Onchocerciasis Technical Advisory Subgroup recently recommended that, for programmatic decision-making, SD Bioline Ov16 RDTs should be performed on DBS in a laboratory setting rather than on whole blood in the field [20]. In accordance with this recommendation, Ov16 SD Bioline RDTs were performed on DBS from a subset of participants at the University of Antwerp. To do so, DBS were incubated for four hours in a 10-millimolar (mM) Tris-HCl–0.1% Tween buffer at room temperature. Following incubation, 10 microlitres (μl) of the DBS eluate was added to the test cassette followed by four drops of reaction buffer. Test results were recorded after 20, 30 and 60 minutes, with photographs taken at each timepoint.

## Skin snip microscopy and O-150 qPCR

After skin snip collection, the biopsy was incubated for 24 hours in 500μl of saline within a 96-well plate at room temperature. The resulting solution was then examined on-site by a skilled technician for the presence of microfilariae under a 10x microscope, and microfilaria counts were recorded per biopsy. Both the skin snip and the remaining saline solution were stored at -20°C until shipment to Antwerp, where they were transferred to -80°C storage. For molecular analysis, one skin snip per participant along with 100μl of the corresponding saline solution was DNA extracted using the QIAamp DNA mini kit (QIAGEN). The resulting DNA was used to perform the O-150 qPCR following an in-house protocol [21]. Each 20μl PCR sample contained 5μl template DNA, 10μl SsOAdvanced Universal SYBR Green Supermic (Bio-Rad, California,

USA), the 0.5µM of the primers (O-150-F: 5′-TCGCCGTGTAAATGTGGAA-3′; O-150-R: 5′-GATTAGGGTCATAGGTC ATCA GTT-3′) and nuclease-free water. The cycler program starts with a 10-minute 95°C denaturation followed by 45 cycles of the reading stage (95°C, 15 seconds; 49°C, 15 seconds, 60°C, 30 seconds) and a melting curve from 60-90°C with a 0.5°C step increase. A standard curve of O-150 plasmid ($10^8$ -$10^5$) was added to each plate.

## Anti-Ov16 ELISA

The anti-Ov16 ELISA was used to detect IgG4 antibodies in DBS samples against the Ov16 antigen. DBS samples were incubated overnight at 4°C in 300µl of 20% soymilk PBS-Tween20. ELISA plates were coated overnight using 2µg/ml of GTS-tagged Ov16 antigens in a 0.05M carbonate buffer. After washing with a Tween20-TrisHCl-NaCl buffer, plates were blocked using 20% soymilk. Subsequently, after washing, 100µl of 1:300 diluted DBS samples in 20% soymilk were added to the plate in duplicate and incubated for two hours at room temperature. After another washing step, 100µl of 1:2000 diluted HRP-conjugated anti-human antibodies in 20% soymilk was added to the plate and incubated for one hour at room temperature. After incubation, plates were washed and 100µl of SigmaFAST OPD (Sigma-Aldrich, Germany) were added to develop for 20 minutes at 37°C. The reaction was stopped by adding 3M HCl. Optical density was measured at 450 and 492 nanometers (nm).

## *Mansonella* species and *M. perstans* qPCR

To assess the presence of co-infections with *Mansonella* parasites, two qPCRs were performed: one targeting all *Mansonella* spp. and another specific for *Mansonella perstans*. Primers and probes described by Bassene *et al*. [22] were used, targeting the internal transcribed spacer 1 (ITS1) of the nuclear ribosomal gene of the parasite. DNA used for these qPCRs was extracted from DBS using the QIAamp DNA mini kit (QIAGEN). Each 20µl PCR sample contains 5µl template DNA, 10µl TaqMan Fast Advanced Master Mix (Life Technologies, Thermo Scientific), 0.2µM of the primers (*Mansonella spp.* qPCR: Mans-F: 5′-CCTGCGGAAGGATCATTAAC-3′, Mans-R: 5′-ATCGACGGTTTAGGCGATAA-3′; *M. perstans* qPCR: Pers-F: 5′-AGGATCATTAACGAGCTTCC-3′; Pers-F: 5′-CGAATATCACCGTTAATTCAGT-3′), 0.2µM of the probes (*Mansonella spp.* qPCR: Mans-P: 6-FAM-CGGTGATATTCGTT GGTGTCT-TAMRA; *M. perstans* qPCR: Pers-P: 6-FAM-TTCACTTTTATTTAGCAACATGCA-TAMRA), and nuclease-free water. The qPCR cycling protocol included an initial denaturation step at 95°C for 2 minutes, followed by 45 cycles of amplification (95°C, 15 seconds; 60°C, 45 seconds). Each plate included a historical, microscopy-positive sample as positive control.

## Data collection and analysis

Field data was uploaded from REDCap, exported, and cleaned in Excel before analysis in R (v4.2.2). Two researchers analysed all photos, with a third resolving disagreements. Photos taken outside a five-minute window were discarded. Percentage of positives were calculated per test, and differences in *O. volvulus* seroprevalence were tested using the "prop.test()" function. The total number of tests included in the analysis differed due to temporary test stockouts and/or availability of samples.

Diagnostic performance calculations were performed using five different 'gold standard' assumptions to account for the absence of a universally accepted reference standard to evaluate *O. volvulus* exposure. These assumptions included three individual reference tests (anti-Ov16 ELISA, O-150 qPCR, skin snip microscopy), and two composite reference standards. Anti-Ov16 ELISA was used as a reference standard for *O. volvulus* exposure, while O-150 qPCR detecting *O. volvulus* DNA and skin snip microscopy were considered reference standards for active infection. The first composite reference standard classified a sample positive if any of the reference standards were positive. The second composite reference defined a sample positive if (a) the microscopy was positive or (b) both O-150 qPCR and anti-Ov16 ELISA were positive. Based on these assumptions, the performance of each RDT was calculated for (i) test read by HCWs, (ii) results

obtained from photographs taken at 30 minutes and (iii) Ov16 SD Bioline test performed in the laboratory on DBS. Additionally, performance was assessed separately for each individual test line of the biplex RDTs.

## Results

A total of 319 women were enrolled in the study between June 22 and September 15, 2023. All women were either pregnant or at most one-week post-partum. Among them, 204 (63.9%) were recruited from the prenatal clinic, 105 (32.9%) from the maternity or post-delivery ward, and one from the vaccination centre. For nine participants, the enrolment location was not recorded. Ivermectin intake history was available for 304 (95.3%) participants, among them 255 (83.9%) reported having taken ivermectin at least once in the past and 15 (5.9%) had received it during the most recent distribution round in February 2023.

### *O. volvulus* prevalence estimates based on reference tests

Anti-Ov16 ELISA detected *O. volvulus* antibodies in 49.6% of participants. *O. volvulus* infection was documented by skin snip microscopy in 31.9% of participants and in 41.8% by qPCR (Table 1).

Among the 43 qPCR-positive individuals, 16 (37.2%) tested negative for the anti-Ov16 ELISA. Only 15 (37.5%) of the 40 participants with microfilariae in their skin snips were also O-150 qPCR positive. Out of the 15 women who took ivermectin during the last CDTI round in February 2023, nine were skin snipped, of which only one tested positive by microscopy and five tested positive by O-150 qPCR.

### *Mansonella* endemicity

The overall prevalence of *Mansonella* spp. was 13.4% (34/253; 95%CI: 9.6-18.4), with *M. perstans* specifically identified in 11.9% of participants (30/251; 8.5-16.9). Based on the O-150 qPCR reference standard, co-infection with *O. volvulus* and *Mansonella* spp. was observed in 7.3% (7/96; 95%CI: 3.2-14.9%) of participants and 5.1% (5/99; 95%CI: 1.9-11.9%) when using skin snip microscopy. Notably, 3 of the 15 (20%) skin snip-positive and O-150 qPCR-negative samples were *Mansonella* spp. positive.

### *O. volvulus* prevalence estimates based on rapid diagnostic tests

When tested in the field by HCWs, significantly higher proportions of *O. volvulus* antibody postitives were observed with the RDT monoplexes (49–65%) compared to the biplexes (34–50%) ($p < 0.001$). However, no difference was observed among the monoplexes ($p = 0.83$) nor among the DDTD types ($p = 0.26$) (Table 2).

Of the 1036 RDT photographs taken by HCWs, only 574 (55.4%) were taken at the right time (within a 5-minute window) and were valid for analyses (not too blurry and containing valid ID and RTD information). The proportion of *O. volvulus* antibody positives differed significantly between RDT monoplexes and biplexes ($p < 0.001$) but not among the different monoplex types ($p = 0.83$) nor among the DDTD types ($p = 0.61$) (Table 2). In general, the proportion of positives on photo interpretation was slightly but not significantly lower than those recorded by HCWs, except for the GADx test (53.4% by photo-based reading versus 62.9% by HCW reading at 30 minutes; $p = 0.02$). No significant differences in positivity were observed across different photo time points.

**Table 1. *O. volvulus* prevalence based on anti-Ov16 ELISA, skin snip microscopy, and O-150 qPCR.**

| Reference test | Prevalence % (#/N; 95%CI) |
| --- | --- |
| Anti-Ov16 ELISA | 49.6 (123/248; 43.2-56.0) |
| Skin snip microscopy | 31.9 (46/144; 24.6-40.3) |
| O-150 qPCR | 41.8 (59/141; 33.7-50.5) |

95%CI = 95% Confidence interval; # = positives; N = total performed.

**Table 2.** *O. volvulus* antibody prevalence derived from RDT results read by healthcare workers, obtained by photo and laboratory testing of dry blood spot.

| Type | Read-out time in min | RDT | T1 | T2 | T1&T2 | NV | Total | Prevalence % (95%CI) |
|------|------|------|------|------|------|------|------|------|
| HCWs | 30 | SD Bioline | 179 | NA | NA | 0 | 290 | 61.7 (55.8-67.3) |
| | | GADx | 185 | NA | NA | 1 | 294 | 62.9 (57.1-68.4) |
| | | DDTD A | 46 | 14 | 121 | 0 | 244 | 49.6 (43.2-56.0) |
| | | DDTD C | 48 | 10 | 125 | 0 | 252 | 49.6 (43.3-56.0) |
| Photo | 20 | GADx | 91 | NA | NA | 0 | 181 | 50.3 (42.8-57.7) |
| | | SD BIOLINE | 94 | NA | NA | 0 | 185 | 50.8 (43.4-58.2) |
| | | DDTD A | 28 | 12 | 46 | 0 | 136 | 33.8 (26.1-42.5) |
| | | DDTD C | 41 | 10 | 54 | 1 | 168 | 32.1 (25.3-39.8) |
| | 30 | GADx | 95 | NA | NA | 0 | 178 | 53.4 (45.8-60.8) |
| | | SD BIOLINE | 89 | NA | NA | 0 | 182 | 48.9 (41.5-56.4) |
| | | DDTD A | 24 | 12 | 55 | 0 | 136 | 40.4 (32.2-49.2) |
| | | DDTD C | 33 | 9 | 66 | 0 | 169 | 39.1 (31.7-46.9) |
| | 60 | GADx | 103 | NA | NA | 0 | 200 | 51.5 (44.4-58.6) |
| | | SD BIOLINE | 102 | NA | NA | 0 | 200 | 51.0 (43.9-58.1) |
| | | DDTD A | 37 | 11 | 65 | 1 | 156 | 41.7 (33.9-49.8) |
| | | DDTD C | 42 | 10 | 71 | 2 | 191 | 37.2 (30.4-44.5) |
| Laboratory | 20 | SD BIOLINE | 58 | NA | NA | 0 | 95 | 61.1 (50.5-70.7) |
| | 30 | SD BIOLINE | 62 | NA | NA | 0 | 95 | 65.3 (54.7-74.5) |
| | 60 | SD BIOLINE | 62 | NA | NA | 0 | 95 | 65.3 (54.7-74.5) |

A DDTD biplex RTD was considered positive when both the test line 1 and 2 were visible. HCW = Health care worker; min = minutes; T1 = Test line 1; T2 = Test line 2; NV = non-valid; RDT = Rapid Diagnostic Test; 95%CI = 95% Confidence interval; NA = Not applicable.

The seroprevalence derived from the Ov16 SD Bioline RDT performed in the laboratory did not significantly differ from the Ov16 SD Bioline RDT results derived from the HCW data (61.7%, 95%CI: 55.8-67.3%; p = 0.62), though they were significantly higher than the Ov16 SD Bioline RDT seroprevalence recorded by photo at 30 minutes (48.9%, 95%CI: 41.5-56.4%; p = 0.03).

**Diagnostic performance of RDT's against reference standards (Table 3)**

The commercially available Ov16 SD Bioline RDT demonstrated sensitivities between 58.3% and 90.0% (mean: 69.4%), with its highest values recorded in laboratory testing on dry blood spots. Sensitivity of DDTD A ranged from 33.7% to 67.7% (mean: 48.9%), and sensitivity of DDTD C from 37.5% to 59.4% (mean: 49.9%).

In all evaluations using the HCWs data, the GADx RDT met or exceeded the WHO's 60% sensitivity threshold for onchocerciasis elimination mapping. Using results recorded by photo, GADx still met this threshold, except for two reference standards: O-150 qPCR and the "any" composite reference (range from 51.6% to 74.1%, mean: 64.5%). None of the RDTs reached the 99.8% specificity threshold set by WHO. Among the four tests, DDTD C reached the highest estimated specificity at 73.2% (mean: 51.4%, min: 42.9%), followed by DDTD A (mean: 56.3%, min: 42.4%, max: 69.7%), GADx (mean: 47.3%, min: 37.5%, max: 65.1%), and lastly SD Bioline (mean: 42.5%, min: 20.0%, max: 62.5%).

**Diagnostic performance of DDTD RDTs according to the visibility of individual test lines, compared against reference standards (Table 4)**

Test line 1 of DDTD type A and C achieved mean sensitivities of 68.9% and 73.0%, respectively, while test line 2 of type A reached 74.6%, meeting the WHO's 60% sensitivity threshold for onchocerciasis elimination mapping. In contrast, test line 2 of type C had a mean sensitivity of 53.4%. When a biplex result was considered positive if either line (T1 or T2)

**Table 3. Diagnostic performance of Rapid Diagnostic Tests against reference standards.**

| Reference test | RDT | True pos. | False pos. | True neg. | False neg. | Sensitivity | Specificity |
|---|---|---|---|---|---|---|---|
| Anti-Ov16 ELISA | SD BIOLINE HCW | 80 | 64 | 50 | 35 | 69.6* | 43.9 |
| | GADx HCW | 75 | 70 | 47 | 40 | 65.2* | 40.2 |
| | DDTD A HCW | 61 | 43 | 58 | 47 | 56.5 | 57.4 |
| | DDTD C HCW | 58 | 43 | 59 | 46 | 55.8 | 57.8 |
| | SD BIOLINE PHOTO30 | 36 | 35 | 27 | 22 | 62.1* | 43.6 |
| | GADX PHOTO30 | 36 | 31 | 33 | 22 | 62.1* | 51.6 |
| | DDTD A PHOTO30 | 29 | 17 | 39 | 26 | 52.7 | 69.6 |
| | DDTD C PHOTO30 | 31 | 21 | 40 | 26 | 54.4 | 65.6 |
| | SD BIOLINE LAB 30 | 30 | 28 | 18 | 5 | 85.7* | 39.1 |
| Microscopy | SD BIOLINE HCW | 31 | 51 | 35 | 11 | 73.8* | 40.7 |
| | GADx HCW | 31 | 55 | 36 | 11 | 73.8* | 39.6 |
| | DDTD A HCW | 21 | 30 | 41 | 10 | 67.7* | 57.8 |
| | DDTD C HCW | 19 | 36 | 39 | 13 | 59.4 | 52.0 |
| | SD BIOLINE PHOTO30 | 21 | 18 | 25 | 6 | 77.8* | 58.1 |
| | GADX PHOTO30 | 19 | 15 | 28 | 8 | 70.4* | 65.1 |
| | DDTD A PHOTO30 | 9 | 10 | 23 | 10 | 47.4 | 69.7 |
| | DDTD C PHOTO30 | 12 | 11 | 30 | 10 | 54.5 | 73.2 |
| | SD BIOLINE LAB 30 | 9 | 12 | 9 | 1 | 90.0** | 42.9 |
| O-150 qPCR | SD BIOLINE HCW | 31 | 47 | 29 | 19 | 62.0* | 38.2 |
| | GADx HCW | 35 | 47 | 31 | 17 | 67.3* | 39.7 |
| | DDTD A HCW | 18 | 34 | 25 | 25 | 41.9 | 42.4 |
| | DDTD C HCW | 20 | 34 | 28 | 24 | 45.5 | 45.2 |
| | SD BIOLINE PHOTO30 | 14 | 25 | 21 | 10 | 58.3 | 45.7 |
| | GADX PHOTO30 | 13 | 24 | 22 | 11 | 54.2 | 47.8 |
| | DDTD A PHOTO30 | 8 | 15 | 18 | 14 | 36.4 | 54.6 |
| | DDTD C PHOTO30 | 9 | 17 | 23 | 14 | 39.1 | 57.5 |
| | SD BIOLINE LAB 30 | 3 | 16 | 9 | 2 | 60.0* | 36.0 |
| Composite reference ANY | SD BIOLINE HCW | 100 | 11 | 5 | 70 | 58.8 | 31.3 |
| | GADx HCW | 110 | 8 | 8 | 63 | 63.6* | 50.0 |
| | DDTD A HCW | 68 | 8 | 7 | 75 | 47.5 | 46.7 |
| | DDTD C HCW | 69 | 8 | 6 | 78 | 46.9 | 42.9 |
| | SD BIOLINE PHOTO30 | 55 | 3 | 5 | 38 | 59.1 | 62.5 |
| | GADX PHOTO30 | 49 | 5 | 3 | 46 | 51.6 | 37.5 |
| | DDTD A PHOTO30 | 27 | 4 | 3 | 53 | 33.8 | 42.9 |
| | DDTD C PHOTO30 | 33 | 4 | 4 | 55 | 37.5 | 50.0 |
| | SD BIOLINE LAB 30 | 35 | 4 | 1 | 18 | 66.0* | 20.0 |
| Composite reference OR | SD BIOLINE HCW | 37 | 37 | 27 | 15 | 71.2* | 42.2 |
| | GADx HCW | 40 | 38 | 29 | 14 | 74.1* | 43.3 |
| | DDTD A HCW | 25 | 22 | 29 | 17 | 59.5 | 56.9 |
| | DDTD C HCW | 23 | 26 | 28 | 20 | 53.5 | 51.9 |
| | SD BIOLINE PHOTO30 | 23 | 14 | 17 | 9 | 71.9* | 54.8 |
| | GADX PHOTO30 | 20 | 13 | 18 | 12 | 62.5* | 58.1 |
| | DDTD A PHOTO30 | 11 | 8 | 15 | 13 | 45.8 | 65.2 |
| | DDTD C PHOTO30 | 14 | 9 | 20 | 13 | 51.9 | 69.0 |
| | SD BIOLINE LAB 30 | 9 | 11 | 7 | 3 | 75.0* | 38.9 |

*(Continued)*

**Table 3.** (Continued)

A DDTD biplex RTD was considered positive if both test line 1 and 2 were visible; Composite reference test ANY is considered positive if any reference test is positive; Composite reference OR is considered positive if microscopy is positive or both anti-Ov16 ELISA and O-150 qPCR are positive; RDT = Rapid Diagnostic Test; pos. = positive; neg. = negative; PPV = positive predictive value; NPV = negative predictive value; * = Target product profile threshold of ≥60% for onchocerciasis elimination mapping; ** = Target product profile threshold of ≥89% for stop-MDA decisions.

was visible, the mean sensitivity increased (type A: 74.6%, type C: 76.6%). The highest sensitivities were observed using microscopy as the reference and T1 or T2 visible, 93.6% for type A and 90.6% for type C, meeting WHO's 89.9% sensitivity threshold for stop-MDA decisions. However, despite these gains in sensitivity, none of the RDTs met the WHO's 99.8% specificity threshold. The average specificities were 41.2% for type A and 40.8% for type C.

## Discussion

A high seroprevalence of onchocerciasis was observed among pregnant and post-partum women in Maridi, with approximately 60% testing positive for *O. volvulus* antibodies using monoplex RDTs (SD Bioline and GADx). In contrast, seroprevalence was about 50% with the biplex RDTs (DDTD A and C). A large proportion of participants had active infections, as microfilariae were observed by microscopy in 31.9% of skin snips, and *O. volvulus* DNA in 41.8% of samples using O-150 qPCR. Additionally, *Mansonella* spp. infection was observed in 13.4% of participants, with 7.3% co-infected with *O. volvulus*. The high prevalence of *O. volvulus* infection may be explained by Maridi's history as a high transmission area and because the vast majority of study participants had not taken ivermectin in the six months prior to skin snip testing due to their pregnancy [23].

Similar to the commercially available Ov16 SD Bioline RDT, none of the novel RDTs reached the WHO specificity threshold of 99.8%. Nevertheless, DDTD type C and A demonstrated higher specificity than the Ov16 SD Bioline RDT. The GADx RDT showed only marginal improvement in specificity over the Ov16 SD Bioline RDT but exhibited a more balanced overall performance (sensitivity/specificity ratio) compared to the DDTD RDTs.

When interpreting test performance, it is essential to consider that different diagnostic methods target distinct parasite-human interactions. Skin snip microscopy and O-150 qPCR detect active infections by identifying microfilariae in the skin, whereas RDTs and ELISAs detect exposure by measuring immune responses against the parasite. Exposure differs from active infection, as it reflects the immune system's 'memory' of past or current infections, or even exposure to L3 larvae from blackflies without necessarily developing an active infection.

Additionally, ivermectin temporarily reduces the microfilaria load in the skin by killing microfilariae and suppressing their production by adult worms for several months, thereby decreasing the apparent prevalence of active infection [24]. However, since ivermectin is contraindicated during pregnancy, recent treatment is unlikely to have affected the study population. Nevertheless, ivermectin distribution has been ongoing in Maridi since the early 2000s, which may have increased the likelihood of false positives when evaluating RDTs' performance against active infection indicators. Long-term exposure to *O. volvulus* can lead to sustained antibody responses in individuals who are no longer actively infected, thereby reducing the specificity estimates for RDTs when calculated with references detecting active infection.

While the monoplexes achieved higher sensitivity percentages, the biplexes showed superior specificity. Both monoplexes reached the 60% threshold for sensitivity required for onchocerciasis elimination mapping. In contrast the DDTD A and C biplexes did not reach this threshold when applying the manufacturer's positivity criterion (requiring both T1 and T2 lines visible). However, with an alternative positivity criterion (T1 or T2 line visible) the DDTD A and C RDTs exceeded the 60% sensitivity, with values reaching up to 93.55%. This increased sensitivity, however, typically comes at the expense of specificity, which explains the manufacturer's more stringent two-line readout. Depending on the objectives of future studies, alternative positivity criteria may be considered to balance sensitivity and specificity according to programmatic needs.

**Table 4. Diagnostic performance of DDTD RDTs based on different positivity criteria, compared against reference standards.**

| Positivity Criteria | Reference test | RDT | True pos. | False pos. | True neg. | False neg. | Sensitivity | Specificity |
|---|---|---|---|---|---|---|---|---|
| T1 or T2 | Anti-Ov16 ELISA | DDTD A HCW | 88 | 67 | 34 | 20 | 81.5* | 33.7 |
| | | DDTD C HCW | 87 | 64 | 38 | 17 | 83.7* | 37.3 |
| | | DDTD A PHOTO30 | 42 | 33 | 23 | 13 | 76.4* | 41.1 |
| | | DDTD C PHOTO30 | 44 | 39 | 22 | 13 | 77.2* | 36.1 |
| | Microscopy | DDTD A HCW | 30 | 45 | 14 | 13 | 93.6** | 35.2 |
| | | DDTD C HCW | 30 | 48 | 14 | 14 | 90.6** | 33.3 |
| | | DDTD A PHOTO30 | 13 | 23 | 10 | 9 | 79.0* | 48.5 |
| | | DDTD C PHOTO30 | 15 | 28 | 12 | 8 | 86.4* | 48.8 |
| | O-150 qPCR | DDTD A HCW | 29 | 46 | 25 | 2 | 69.8* | 23.7 |
| | | DDTD C HCW | 29 | 50 | 25 | 3 | 68.2* | 22.6 |
| | | DDTD A PHOTO30 | 15 | 17 | 16 | 4 | 59.1 | 30.3 |
| | | DDTD C PHOTO30 | 19 | 21 | 20 | 3 | 65.2* | 30.0 |
| | Composite reference ANY | DDTD A HCW | 100 | 11 | 4 | 43 | 69.9* | 26.7 |
| | | DDTD C HCW | 100 | 11 | 3 | 47 | 68.0* | 21.4 |
| | | DDTD A PHOTO30 | 48 | 6 | 1 | 32 | 60.0* | 14.3 |
| | | DDTD C PHOTO30 | 59 | 5 | 3 | 29 | 67.1* | 37.5 |
| | Composite reference OR | DDTD A HCW | 36 | 34 | 17 | 6 | 85.7* | 33.3 |
| | | DDTD C HCW | 35 | 37 | 17 | 8 | 81.4* | 31.5 |
| | | DDTD A PHOTO30 | 17 | 14 | 9 | 7 | 70.8* | 39.1 |
| | | DDTD C PHOTO30 | 21 | 17 | 12 | 6 | 77.8* | 41.4 |
| T1 | Anti-Ov16 ELISA | DDTD A HCW | 81 | 63 | 38 | 27 | 75.0* | 37.6 |
| | | DDTD C HCW | 82 | 61 | 41 | 22 | 78.9* | 40.2 |
| | | DDTD A PHOTO30 | 37 | 29 | 27 | 18 | 67.3* | 48.2 |
| | | DDTD C PHOTO30 | 40 | 36 | 25 | 17 | 70.2* | 41.0 |
| | Microscopy | DDTD A HCW | 27 | 40 | 19 | 16 | 83.9* | 40.9 |
| | | DDTD C HCW | 28 | 45 | 17 | 16 | 87.5* | 37.3 |
| | | DDTD A PHOTO30 | 12 | 20 | 13 | 10 | 79.0* | 57.6 |
| | | DDTD C PHOTO30 | 14 | 25 | 15 | 9 | 86.4* | 56.1 |
| | O-150 qPCR | DDTD A HCW | 26 | 42 | 29 | 5 | 62.8* | 32.2 |
| | | DDTD C HCW | 28 | 47 | 28 | 4 | 63.6* | 27.4 |
| | | DDTD A PHOTO30 | 15 | 14 | 19 | 4 | 54.6 | 39.4 |
| | | DDTD C PHOTO30 | 19 | 18 | 23 | 3 | 60.9* | 37.5 |
| | Composite reference ANY | DDTD A HCW | 94 | 9 | 6 | 49 | 65.7* | 40.0 |
| | | DDTD C HCW | 96 | 10 | 4 | 51 | 65.3* | 28.6 |
| | | DDTD A PHOTO30 | 43 | 5 | 2 | 37 | 53.8 | 28.6 |
| | | DDTD C PHOTO30 | 55 | 4 | 4 | 33 | 62.5* | 50.0 |
| | Composite reference OR | DDTD A HCW | 32 | 31 | 20 | 10 | 76.2* | 39.2 |
| | | DDTD C HCW | 33 | 35 | 19 | 10 | 76.7* | 35.2 |
| | | DDTD A PHOTO30 | 17 | 11 | 12 | 7 | 70.8* | 52.2 |
| | | DDTD C PHOTO30 | 21 | 14 | 15 | 6 | 77.8* | 51.7 |
| T2 | Anti-Ov16 ELISA | DDTD A HCW | 68 | 47 | 54 | 40 | 63.0* | 53.5 |
| | | DDTD C HCW | 63 | 46 | 56 | 41 | 60.6* | 54.9 |
| | | DDTD A PHOTO30 | 34 | 21 | 35 | 21 | 61.8* | 62.5 |
| | | DDTD C PHOTO30 | 35 | 24 | 37 | 22 | 61.4* | 60.7 |

*(Continued)*

**Table 4.** (Continued)

| Positivity Criteria | Reference test | RDT | True pos. | False pos. | True neg. | False neg. | Sensitivity | Specificity |
|---|---|---|---|---|---|---|---|---|
| | Microscopy | DDTD A HCW | 21 | 39 | 20 | 22 | 77.4* | 52.1 |
| | | DDTD C HCW | 22 | 37 | 25 | 22 | 62.5 | 48.0 |
| | | DDTD A PHOTO30 | 9 | 18 | 15 | 13 | 47.4 | 60.6 |
| | | DDTD C PHOTO30 | 10 | 20 | 20 | 13 | 54.6 | 65.9 |
| | O-150 qPCR | DDTD A HCW | 24 | 34 | 37 | 7 | 48.8 | 33.9 |
| | | DDTD C HCW | 20 | 39 | 36 | 12 | 50.0 | 40.3 |
| | | DDTD A PHOTO30 | 9 | 13 | 20 | 10 | 40.9 | 45.5 |
| | | DDTD C PHOTO30 | 12 | 14 | 27 | 10 | 43.5 | 50.0 |
| | Composite reference ANY | DDTD A HCW | 74 | 10 | 5 | 69 | 51.8 | 33.3 |
| | | DDTD C HCW | 73 | 9 | 5 | 74 | 49.7 | 35.7 |
| | | DDTD A PHOTO30 | 32 | 5 | 2 | 48 | 40.0 | 28.6 |
| | | DDTD C PHOTO30 | 37 | 5 | 3 | 51 | 42.1 | 37.5 |
| | Composite reference OR | DDTD A HCW | 29 | 25 | 26 | 13 | 69.1* | 51.0 |
| | | DDTD C HCW | 25 | 28 | 26 | 18 | 58.1 | 48.2 |
| | | DDTD A PHOTO30 | 11 | 11 | 12 | 13 | 45.8 | 52.2 |
| | | DDTD C PHOTO30 | 14 | 12 | 17 | 13 | 51.9 | 58.6 |

Different DDTD positivity criteria used were: T1 = Test is positive if line 1 is visible; T2 = Test is positive if line 2 is visible; T1 or T2 = Any line visible for the test to be positive; Composite reference test ANY is considered positive if any reference test is positive; Composite reference OR is considered positive if microscopy is positive or both anti-Ov16 ELISA and O-150 qPCR are positive; RDT = Rapid Diagnostic Test; pos. = positive; neg. = negative; PPV = positive predictive value; NPV = negative predictive value; * = Target product profile threshold of ≥60%; ** = Target product profile threshold of ≥89%.

We also tested stored DBS from a subset of participants using the Ov16 SD Bioline RDT in the laboratory, as recommended by WHO [20]. This approach significantly increased the test's sensitivity compared to the Ov16 SD Bioline RDT conducted in the field —whether interpreted by HCWs or by photo-based readings—, but at the cost of reduced specificity. The improved sensitivity reflects the controlled laboratory conditions, where consistent lighting and singular task focus contributed to more accurate readings. In contrast, HCWs performed the tests while managing their primary duties as midwives, often in poorly lit settings, which may have affected result interpretation. Further studies should explore whether the use of extra light sources and/or standardised photo-based reading (potentially app-assisted) could improve the performance of RDTs in field conditions. Similar laboratory-field comparisons should be considered for the novel RDTs.

Our study has several limitations. Skin snip testing was performed in only 56% of the participants, partly because some women gave birth on the same day of their enrolment. These women were often bedridden for one or several days after labour, making it difficult or impossible for them to visit the laboratory for skin snip testing. Another contributing factor is that skin snip collection is an invasive procedure, which is often perceived by communities as painful and unacceptable [5]. Additionally, 14.1% of the photos were taken outside the recommended five-minute window, suggesting that a comparable percentage of RDTs may have been read by HCWs outside of the correct 30-minute interpretation time. This highlights the importance of HCW training and test stability over time, especially when integrating RDTs into the routine activities of HCWs. Furthermore, 14.2% of the tests were classified as blurry or unreadable, further reducing the usable dataset. To improve photo retention and quality in future studies, a dedicated RDT photo station should be considered, equipped with a fixed camera and a standardized setup to ensure consistent positioning of RDTs within the frame.

The disagreement between O-150 qPCR and skin snip microscopy results observed in the study is concerning. Only 15 (37.5%) of the 40 participants with microfilariae detected by microscopy tested positive by O-150 qPCR. This discrepancy may be explained by limited specificity of microscopy, the use of only one skin snip for DNA extraction, and/or suboptimal

qPCR design. However, results from the 9 women who took ivermectin during the last round of CDTI and were skin snipped, suggest an opposite pattern: five (55.6%) were 0–150 qPCR positive while none had microfilaria by microscopy indicating that O-150 qPCR may in fact be more sensitive. On the other hand, of the 25 participants microscopy-positive but O-150 qPCR-negative, three tested positive for *Mansonella* spp., suggesting that some microscopy-positive results may have been due to misidentifications. *M. streptocerca* microfilariae, which reside in the skin, could have been mistaken for *O. volvulus*, while deeper skin snips that inadvertently included blood may have contained *M. perstans parasites*. Differentiating filarial species by microscopy remains challenging, highlighting the importance of considering other filarial species as potential sources of diagnostic error.

## Conclusion

A high (sero)prevalence of onchocerciasis and a moderate *Mansonella spp.* prevalence were observed among pregnant and post-partum women in Maridi. While the study design did not permit a definitive assessment of the sensitivity of the tested RDTs – as it was designed to add to a larger multi-country performance assessment –, estimates based on different reference test suggest sensitivities of the RDTs close to 60%, meeting the WHO's set threshold for onchocerciasis elimination mapping, but show specificities below the 99.8% threshold. The highest specificity was observed with the DDTD type C RDT, but this test scored poorly in sensitivity. The GADx RDT offered a more balanced profile in terms of sensitivity/specificity ratio. Performing the Ov16 SD Bioline Ov16 in the laboratory rather than in the field increased its sensitivity but reduced its specificity. A similar comparison of laboratory versus field-based performance should also be conducted with the novel RDTs. Further research should investigate whether standardised photo-based reading, ideally supported by a mobile app, could improve the RDT performance in field settings. Finally, the suboptimal specificity observed across all tests highlights the urgent need for improved diagnostic tools to support onchocerciasis elimination efforts.

## Acknowledgments

We would like to thank the community, the hospital, and the commissioner of Maridi County for their acceptance and support of this study. Furthermore, we thank all healthcare workers involved during the sample collection.

## Author contributions

**Conceptualization:** Robert Colebunders.

**Data curation:** Amber Hadermann, Stephen Raimon Jada.

**Formal analysis:** Amber Hadermann, Chiara Travisan, Abdurezak Seid, Marina Saleeb.

**Funding acquisition:** Robert Colebunders.

**Investigation:** Amber Hadermann, Stephen Raimon Jada, Chiara Travisan, Abdurezak Seid.

**Methodology:** Amber Hadermann.

**Project administration:** Stephen Raimon Jada.

**Resources:** Robert Colebunders.

**Software:** Amber Hadermann.

**Supervision:** Amber Hadermann, Stephen Raimon Jada, Chiara Travisan, Robert Colebunders.

**Validation:** Amber Hadermann.

**Visualization:** Amber Hadermann.

**Writing – original draft:** Amber Hadermann.

**Writing – review & editing:** Amber Hadermann, Stephen Raimon Jada, Chiara Travisan, Abdurezak Seid, Charlotte Lubbers, Luís-Jorge Amaral, Marina Saleeb, Yak Yak Bol, Katja Polman, Joseph Nelson Siewe Fodjo, Robert Colebunders.

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
